# Differential Associations of Vitamin D Metabolites with Adiposity and Muscle-Related Phenotypes in Korean Adults: Results from KNHANES 2022–2023

**DOI:** 10.3390/nu17183013

**Published:** 2025-09-20

**Authors:** Se-Hong Kim, Yuji Jeong, Seok-Won Son, Ha-Na Kim

**Affiliations:** Department of Family Medicine, St. Vincent’s Hospital, College of Medicine, The Catholic University of Korea, Seoul 06591, Republic of Korea; iron1600@catholic.ac.kr (S.-H.K.); yj.jeong@cmcnu.or.kr (Y.J.); sonsw0826@gmail.com (S.-W.S.)

**Keywords:** 25-hydroxyvitamin D, 25-hydroxyvitamin D_2_, 25-hydroxyvitamin D_3_, 3-epi-25-hydroxyvitamin D_3_, body composition, muscle strength, muscle mass, sarcopenia, obesity

## Abstract

Background: Body composition plays a crucial role in metabolic health, aging, and the development of various diseases. Vitamin D has been implicated in the regulation of muscle and adipose tissue function, but its relationship with body composition remains unclear. Objectives: This study aimed to investigate the association between serum 25-hydroxyvitamin D (25[OH]D) concentration, its subspecies, and body composition parameters and related phenotypes in Korean adults aged ≥20 years. Methods: This cross-sectional study included 2612 eligible participants from the Korea National Health and Nutrition Examination Survey 2022–2023, and data on body composition parameters and serum 25(OH)D levels were analyzed. Serum 25(OH)D levels and subspecies were determined using liquid chromatography-tandem mass spectrometry. Body composition factors included anthropometric indices, muscle strength, and fat and muscle mass, which were assessed using bioelectrical impedance analysis. Results: After multivariable adjustment, serum concentrations of total 25(OH)D and 25(OH)D_3_ were inversely associated with waist circumference, body mass index, and fat mass and positively associated with handgrip strength and lean mass, whereas serum 25(OH)D_2_ and 3-epi-25(OH)D_3_ levels showed no such associations. Higher 25(OH)D_2_ concentrations were positively associated with low muscle strength and the prevalence of sarcopenia. Through non-linear analyses, U-shaped correlations were identified between total 25(OH)D and 25(OH)D_3_ levels with low muscle mass, respectively, while a J-shaped correlation was observed between 25(OH)D_2_ level and low muscle mass. Conclusions: Total 25(OH)D and 25(OH)D_3_ levels were inversely associated with adiposity and positively associated with muscle strength and lean mass, whereas the 25(OH)D_2_ level was linked to low muscle strength and sarcopenia. The U- and J-shaped associations with low muscle mass indicate the complex and differential roles of vitamin D subspecies, warranting further study.

## 1. Introduction

Body composition, defined by the relative amounts of muscle and fat, is fundamental to physiological functioning and healthy aging and has been linked to metabolic disorders, cardiovascular diseases, and mortality. Obesity, characterized by the accumulation of excess body fat, represents a major public health issue because of its association with a higher risk of cardiovascular disease, hypertension, dyslipidemia, and type 2 diabetes mellitus, with its prevalence rising globally [1]. Increased visceral fat plays a critical role in the development of insulin resistance and related metabolic disorders [2], and trunk fat mass is inversely associated with insulin sensitivity [3], whereas leg fat mass has a favorable impact on glucose metabolism and lipid profile [4] and is linked to a lower risk of diabetes [5]. Sarcopenia, characterized by a reduction in muscle mass and strength and/or physical performance, is associated with physical and functional disabilities and increased mortality, making it a significant global health concern. Furthermore, a decrease in muscle mass and strength in the limbs are associated with metabolic disturbances [5,6].

Vitamin D, a prohormone derived from cholesterol, is obtained through both ultraviolet-B exposure and dietary sources [7]. It has been recognized for its classical role in bone metabolism as well as its influence on body composition [8,9]. The vitamin D receptor, present in skeletal muscle and fat tissue, mediates the pleiotropic effects of vitamin D, including the regulation of muscle anabolism, inflammatory signaling, and metabolic function [10,11]. These biological mechanisms indicate that vitamin D status might have significant implications for muscle mass maintenance and fat distribution [9,12,13,14], although the link and mechanism between low vitamin D status and sarcopenia or body fat are not yet fully understood.

Serum 25-hydroxyvitamin D [25(OH)D] concentration, defined as the total level of 25-hydroxyvitamin D_2_ [25(OH)D_2_] and 25-hydroxyvitamin D_3_ [25(OH)D_3_] formed in the liver through metabolic transformations of the prohormone vitamin D, can be used to determine vitamin D status [15]. Therefore, accurate measurement of serum 25(OH)D levels is crucial for ensuring proper diagnosis and effective monitoring of therapeutic response. However, the vitamin D metabolite, 25(OH)D_3_ can also undergo epimerization by the enzyme 3-epimerase, resulting in the formation of 3-epi-25-hydroxyvitamin D_3_ [3-epi-25(OH)D_3_], which shows a decreased binding affinity for the vitamin D receptor and reduced biological effectiveness [16,17]. Furthermore, when determining 25(OH)D levels, 3-epi-25(OH)D may be accidentally measured due to cross-reactivity in clinical laboratory methods, such as immunoassays, which can falsely increase serum 25(OH)D levels [18] and mask actual deficiencies [19].

Although previous studies have investigated the possible correlations between vitamin D levels and components of various body composition, findings remain inconsistent and inconclusive [8,9,20,21,22]. In particular, the precise evaluation of vitamin D levels is challenged by the presence of metabolites such as 3-epi-25(OH)D_3_, which exhibit reduced biological activity and can interfere with conventional immunoassay-based measurements [16,17]. To address this limitation, analytical methods such as liquid chromatography-tandem mass spectrometry (LC-MS/MS), which accurately differentiate 25(OH)D_3_ from its epimers, could provide more reliable measurements [23]. This study aimed to investigate the association between serum 25(OH)D concentrations and its subspecies quantified using LC-MS/MS and body composition parameters and related phenotypes in a sample of Korean adults that represent the national population, utilizing dataset from the Korea National Health and Nutrition Examination Survey (KNHANES).

## 2. Materials and Methods

### 2.1. Study Population and Design

The Korean Centers for Disease Control and Prevention conduct the KNHANES at 3-year intervals to evaluate public health and provide baseline data for the development, establishment, and evaluation of Korean public health policies. Participants in the KNHANES study were all non-institutionalized individuals aged ≥1 year old, and were selected through a stratified, multistage, cluster probability sampling method to ensure that the KNHANES sample was independent, homogeneous, and representative of the national population. Data were obtained through household interviews, anthropometric and biochemical measurements, as well as nutritional assessments. The Institutional Review Board (IRB) of the Korean Centers for Disease Control and Prevention granted approval for the protocols, and informed consent was obtained from all participants at baseline. The IRB at the Catholic University of Korea granted approval for the analyses performed in this study (IRB approval: no. VC25ZISI0091 and date of approval: 16 May 2025).

In this cross-sectional study, we used data from the KNHANES collected in 2022 and 2023, which originally included data on 4739 adults aged ≥20 years, assessed serum vitamin D levels, and excluded participants with incomplete or missing data for key variables (n = 2090), as well as those who were pregnant (n = 37). Consequently, data from 2612 participants were analyzed (Figure 1).

### 2.2. Measurement of Serum 25-Hydroxyvitamin D Concentrations

Serum concentrations of 25(OH)D_2_, 25(OH)D_3_, and 3-epi-25(OH)D_3_ were measured using blood samples obtained from the antecubital vein of each participant following an overnight fast and determined using LC-MS/MS (Triple Quad 4500MD; PerkinElmer, Waltham, MA, USA). Chromatographic separation was performed using the vitamin D reagent kit (MSMS Vitamin D kit; PerkinElmer, Waltham, MA, USA) with kit-supplied mobile phases at a flow rate of 0.15 mL/min under isocratic conditions on a Kinetex XB C18 column (2.1 × 150 mm, 2.6 μm; Phenomenex, Torrance, CA, USA) maintained at 30 °C, with an injection volume of 2 μL. Quantification was performed in multiple reaction monitoring mode with transitions *m*/*z* 607.4 → 298.1 for 25(OH)D_3_, *m*/*z* 607.4 → 298.1 for 3-epi-25(OH)D_3_, *m*/*z* 619.4 → 298.1 for 25(OH)D_2_, and *m*/*z* 610.4 → 301.1 for the internal standard. Calibration curves were established over the concentration ranges of 1.31–130.60 ng/mL for 25(OH)D_3_ and 3-epi-25(OH)D_3_, and 1.37–137.43 ng/mL for 25(OH)D_2_, with correlation coefficients (r). Accuracy and precision were assessed at four QC levels: for 25(OH)D_3_, LLOQ 0.29 ng/mL, low 7.77 ng/mL, medium 32.90 ng/mL, and high 66.12 ng/mL and for 25(OH)D_2_, LLOQ 0.30 ng/mL, low 8.34 ng/mL, medium 34.58 ng/mL, and high 69.33 ng/mL, respectively [24].

The concentration of serum total 25(OH)D was defined by summing 25(OH)D_2_ and 25(OH)D_3_ concentrations [25], and vitamin D deficiency was defined by a total 25(OH)D concentration of less than 20 ng/mL [26].

### 2.3. Measurements of Body Composition Parameters

After fasting overnight, trained examiners assessed each participant’s weight, waist circumference (WC), and height. The body mass index (BMI) was determined by dividing weight (kg) by the square of height in meters (m^2^).

The measurement of fat and lean body mass was conducted using a multi-frequency bioelectrical impedance device (InBody 720; Biospace Co., Seoul, Republic of Korea). Participants stood and held the device handles, placing their palms, thumbs, and feet on the eight tactile electrodes. Impedance values for both arms, the trunk, and both legs were measured at frequencies of 1, 5, 50, 250, 500, and 1000 kHz. Height-adjusted appendicular skeletal muscle mass (ASM) was determined using the equation: [total limb lean mass/(height)^2^ (kg/m^2^)].

To assess skeletal muscle strength, the handgrip strength (HGS) of participants was evaluated by trained examiners using a digital grip strength dynamometer (T.K.K. 5401; Takei Scientific Instruments, Niigata, Japan). This device is capable of measuring up to 100.0 kg of force and features an adjustable grip span. Participants were instructed to sit with their gaze directed forward and their elbow flexed at a 90° angle, then to exert maximum continuous force on the dynamometer until a reading was displayed on the display. HGS was measured twice for each hand in an alternating manner, commencing with the dominant hand. The highest value obtained from the four measurements was recorded as the HGS.

### 2.4. Definition of Body Composition Phenotypes

We evaluated low muscle mass, low muscle strength, sarcopenia, and obesity as body composition phenotypes of the participants. Obesity was defined by a BMI of ≥25 kg/m^2^ [27]. Low skeletal muscle mass was defined by an ASM < 7.0 kg/m^2^ in males or <5.7 kg/m^2^ in females. Low muscle strength was defined by an HGS < 28 kg in males or <18 kg in females [28]. Sarcopenia was defined by the concurrent occurrence of low muscle strength and low skeletal muscle mass [28].

### 2.5. Other Variables

Information was collected through self-reports on variables including age, sex, smoking status, alcohol consumption, physical activity, household income, and medical history of hypertension, diabetes, and dyslipidemia. Cigarette smoking was divided into three groups based on current usage estimates: non-smokers, ex-smokers, and current smokers. The data on alcohol consumption included both the frequency of drinking days and the quantity of alcoholic beverages consumed per drinking day during the year preceding the KNHANES interview. We applied the Korean version for a “standard drink,” defined as any beverage with 10 g of pure alcohol. This is based on alcohol content percentages of 4.5% in beer, 12% in wine, 6% in the Korean traditional drink makgeolli, 21% in Korean soju, and 40% in whisky. Alcohol consumption was categorized into three groups: abstinence (no alcohol consumed in the past year), moderate drinking (fewer than 14 standard drinks per week for men and fewer than seven for women), and heavy drinking (more than 14 standard drinks per week for men and more than seven for women) [29]. Participants who responded affirmatively to the question “Do you exercise regularly enough to make you sweat?” were assigned to the aerobic exercise group. The group that reported engaging in resistance exercises was defined as the resistance exercise group. Household income was categorized into four quartiles using monthly equivalized household income. This was calculated by dividing the total monthly household income by the square root of the household size. The first quartile indicated the group with the lowest income.

Trained interviewers assessed dietary intake by conducting a 24 h recall of all foods and drinks. The assessment of dietary intake was conducted using the food composition tables from the Rural Development Administration along with the nutrient database provided by the Korea Health and Industry Development Institute [30]. The calculation of energy from the intake of carbohydrates, fats, and proteins was performed using standard conversion factors, which convert grams into kilocalories (4 kcal/g for carbohydrates, 9 kcal/g for fats, and 4 kcal/g for proteins). Each macronutrient’s energy contribution was represented as its proportion of the overall energy intake.

### 2.6. Statistical Analysis

We utilized the SAS PROC SURVEY module (version 9.2, SAS Institute, Cary, NC, USA), which incorporates strata, clusters, and weights, to conduct our data analysis in line with a complex sampling design. The characteristics of the study population were analyzed using the independent *t*-test or Wilcoxon rank-sum test for continuous variables and the x^2^ test or Fisher’s exact test for categorical variables. Data are presented as mean ± standard error or percentages. The associations between serum 25(OH)D and subspecies levels and body composition parameters such as WC, BMI, HGS, ASM, and fat and lean mass, were analyzed using multiple linear regression with age, sex, smoking status, alcohol consumption, exercise, household income, medical history of hypertension, diabetes, and dyslipidemia, energy intake, proportion of macronutrient intake, daily fiber intake, and seasonal variation in vitamin D status as covariates. The odds ratios (ORs) of body composition phenotypes (the presence of low muscle strength, low muscle mass, sarcopenia, and obesity) as dependent variables, according to serum vitamin D subspecies concentrations as the independent variables, were evaluated using multiple logistic analyses. Model 1 was adjusted for age and sex; model 2 was adjusted for age, sex, smoking, alcohol consumption, exercise, household income, energy intake, proportion of macronutrient intake, and fiber intake; and model 3 was adjusted for age, sex, smoking, alcohol consumption, exercise, household income, energy intake, proportion of macronutrient intake, fiber intake, medical history of hypertension, diabetes, and dyslipidemia, and seasonal variation in vitamin D status. Potential non-linear relationships between different vitamin D subspecies levels and the prevalence of body composition phenotypes were assessed using restricted cubic spline regression adjusted for all covariates. All statistical analyses were conducted utilizing SAS software (version 9.2, SAS Institute, Cary, NC, USA), with statistical significance determined at a *p*-value of less than 0.05.

## 3. Results

### 3.1. Characteristics of the Study Population According to Vitamin D Status

A total of 2612 participants were included in this study. Table 1 presents the characteristics of the study population categorized by vitamin D status, defined as a total 25(OH)D level of either <20 ng/mL or ≥20 ng/mL. Participants with vitamin D deficiency were more likely to be male; younger; include more heavy drinkers and current smokers; have a higher daily energy intake; include lower proportion of those with dyslipidemia; and include those with higher WC, BMI, HGS, fat mass, lean mass, and ASM values.

### 3.2. Associations Between Serum 25-Hydroxyvitamin D Concentrations and Body Composition Parameters

The linear relationships between total 25(OH)D, its subspecies, and body composition factors are presented in Table 2. After adjusting for covariates, including age, sex, smoking, alcohol consumption, exercise, household income, energy intake, proportion of macronutrient intake, fiber intake, comorbidities, and seasonal variation in vitamin D status, serum concentrations of total 25(OH)D and 25(OH)D_3_ were negatively associated with WC; BMI; and total, trunk, and limb fat mass, and positively associated with HGS and total and limb lean mass, but serum concentrations of 25(OH)D_2_ and 3-epi-25(OH)D_3_ were not associated with body composition parameters.

### 3.3. Associations Between Serum 25-Hydroxyvitamin D Concentrations and Body Composition Phenotype

Serum concentrations of total 25(OH)D and 25(OH)D_3_ were positively associated with the prevalence of low muscle strength, muscle mass, and sarcopenia and negatively associated with obesity; however, these associations were no longer observed after adjustment for all covariates. The serum concentration of 3-epi-25(OH)D_3_ was associated with low muscle strength, muscle mass, and sarcopenia: however, no significant associations were found after adjusting for covariates. With respect to 25(OH)D_2_ concentrations, increased levels of 25(OH)D_2_ were positively associated with low muscle strength and sarcopenia after adjusting for all covariates (adjusted OR 1.279 [95% CI 1.055–1.552] and adjusted OR 1.418 [95% CI 1.162–1.732], respectively) (Table 3).

### 3.4. Non-Linear Relationship Between Serum 25-Hydroxyvitamin D Concentrations and Body Composition Phenotype

As shown in Figure 2, there was a linear association between serum 25(OH)D_2_ levels and the prevalence of low muscle strength and sarcopenia (*p* for overall = 0.025; *p* for non-linearity = 0.740 and *p* for overall = 0.008; *p* for non-linearity = 0.907, respectively). Although the overall association between 25(OH)D_3_ and low muscle strength was not statistically significant (*p* for overall = 0.100), a potential non-linear trend was observed (*p* for non-linearity = 0.038). Regarding the prevalence of low muscle mass, there were U-shaped relationships of total 25(OH)D and 25(OH)D_3_ concentrations (*p* for overall = 0.030; *p* for non-linearity = 0.017 and *p* for overall = 0.013; *p* for non-linearity = 0.006, respectively), and a J-shaped association was identified between 25(OH)D_2_ concentration and low muscle mass (*p* for overall = 0.021; *p* for non-linearity = 0.039).

## 4. Discussion

In this population-based cross-sectional study, data from 2612 participants from the KNHANES 2022–2023 were included to evaluate the associations between serum 25(OH)D concentrations, including those of total 25(OH)D, 25(OH)D_3_, 25(OH)D_2_, and 3-epi-25(OH)D_3_, and various body composition parameters and phenotypes. Serum concentrations of total 25(OH)D and 25(OH)D_3_ were inversely associated with adiposity indices and positively associated with lean body mass and muscle strength. Serum concentrations of 25(OH)D_2_ were significantly associated with increased odds of low muscle strength and sarcopenia. There were non-linear relationships between 25(OH)D subtypes and body composition phenotype, including a U-shaped association for total 25(OH)D and 25(OH)D_3_ with low muscle mass and a J-shaped association between 25(OH)D_2_ and low muscle mass, respectively. These results underscore the heterogeneous associations between 25(OH)D subspecies and body composition-related parameters and outcomes.

The serum concentration of total 25(OH)D, primarily composed of 25(OH)D_3_, was associated with body composition parameters, which were negatively associated with fat-related factors and positively associated with muscle-related factors, similar to previous studies [31,32,33]. However, ASM, which is used to define low muscle mass in sarcopenia, was not linearly associated with total 25(OH)D concentrations, whereas a U-shaped dose–response association was identified between serum 25(OH)D level and low muscle mass. These results suggest that both vitamin D deficiency and excessive status levels need to be considered for maintaining skeletal muscle mass. Previous research on excessive vitamin D status and muscle health remains limited [31]; therefore, additional research focusing on the impact of excessive serum 25(OH)D concentrations on muscle health is warranted.

Serum 25(OH)D_2_ originates from the hepatic hydroxylation of vitamin D_2_, which is obtained mainly from plant-based sources and fortified foods or supplementation. Compared to 25(OH)D_3_, 25(OH)D_2_ has a shorter half-life, lower affinity for vitamin D-binding protein, and may be less effective in increasing serum 25(OH)D levels [34], but there is a paucity of investigation on the comparison of their impacts on body composition factors and phenotypes. In this study, elevated serum 25(OH)D_2_ concentrations were associated with low muscle strength and a high prevalence of sarcopenia. In previous studies, vitamin D_2_ supplementation had inconsistent effects on muscular function and exercise-induced muscle damage, and there was an increase in serum 25(OH)D_2_ and a decrease in serum 25(OH)D_3_ levels [35,36], implying that adding vitamin D_2_ to the body’s total vitamin D pool could potentially lower the relative concentration of vitamin D_3_. Furthermore, enzymes such as CYP24A1 and CYP3A4, which are involved in the inactivation of vitamin D, may degrade vitamin D_2_ more rapidly than vitamin D_3_, thereby restricting the effectiveness of vitamin D_2_ in the cells where it is present [37]. Taken together, these characteristics—including the shorter half-life of 25(OH)D_2_, its weaker binding to vitamin D-binding protein, and its faster degradation by catabolic enzymes—suggest potential mechanisms by which serum 25(OH)D_2_ may contribute to negative effects on muscle health, thereby making it less effective and possibly detrimental compared to 25(OH)D_3_ for body composition–related outcomes. Additional research is required to determine its efficacy and clinical relevance.

The level of 3-epi-25(OH)D_3_, which is primarily formed through isomerization catalyzed by an epimerase enzyme, can only be quantified using LC-MS/MS [38], which effectively separates epimeric and isobaric interferences chromatographically, allowing for the sensitive and accurate detection of vitamin D metabolites in serum. In contrast, most existing immunoassay techniques produce inaccurate vitamin D measurements because of the interference caused by epimers [18]. Several previous studies have indicated that the proportion of total vitamin D present as 3-epi-25(OH)D_3_ is higher in newborns, infants, and pregnant women than in non-pregnant adults [18,38,39]. Furthermore, although some studies have demonstrated the biological activity of 3-epi-25(OH)D_3_ in vitro, evidence from in vivo and clinical studies remains limited and inconsistent [40,41,42,43], suggesting that 3-epi-25(OH)D_3_ might have important implications for future clinical research. However, in this study, no association was observed between 3-epi-25(OH)D_3_ level and various body composition parameters and phenotypes in Korean adults aged ≥ 20 years.

The strength of the present study in the South Korean population was that it was based on data collected through a comprehensive nationwide survey, ensuring representativeness. Furthermore, this is the first study among Korean adults to examine the associations between serum 25(OH)D concentrations, including subspecies—25(OH)D_2_, 25(OH)D_3_, and 3-epi-25(OH)D_3_—using the LC-MS/MS method and various body composition parameters and related phenotypes, after adjusting for covariates that could affect body composition factors, including physical activities and macronutrient intake. However, this study has several limitations. First, its cross-sectional design precludes causal inference; therefore, the findings should be regarded as hypothesis-generating, underscoring the need for well-designed cohort and intervention studies to validate these associations. Second, detailed information on vitamin D supplementation (e.g., use, dosage, and formulation) was not available, which can have influenced serum 25(OH)D_2_ or 25(OH)D_3_ concentrations. Third, the KNHANES does not include data on physical performance measures, which are a component in the diagnosis of sarcopenia. Finally, as this study was limited to Korean adults, the generalizability of the findings to other ethnic populations remains uncertain.

## 5. Conclusions

Serum concentrations of total 25(OH)D and 25(OH)D_3_ were negatively associated with WC, BMI, and fat mass, and positively associated with HGS and total and limb lean mass in adults aged ≥ 20 years. Higher 25(OH)D_2_ concentrations were positively associated with low muscle strength and the prevalence of sarcopenia. U-shaped correlations were identified between total 25(OH)D and 25(OH)D_3_ with low muscle mass, whereas a J-shaped correlation was observed between 25(OH)D_2_ and low muscle mass. Further studies are needed to elucidate the diverse associations between serum 25(OH)D and its subspecies and various parameters and outcomes related to body composition.

## Figures and Tables

**Figure 1 nutrients-17-03013-f001:**
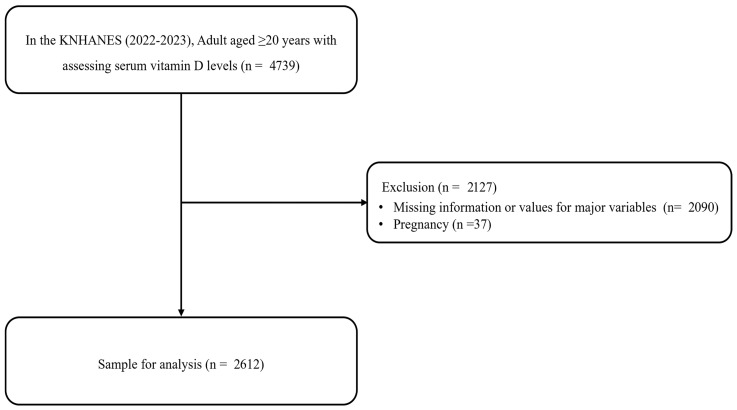
Study population: Data from the 2022–2023 Korean National Health and Nutrition Examination Survey (KNHANES).

**Figure 2 nutrients-17-03013-f002:**
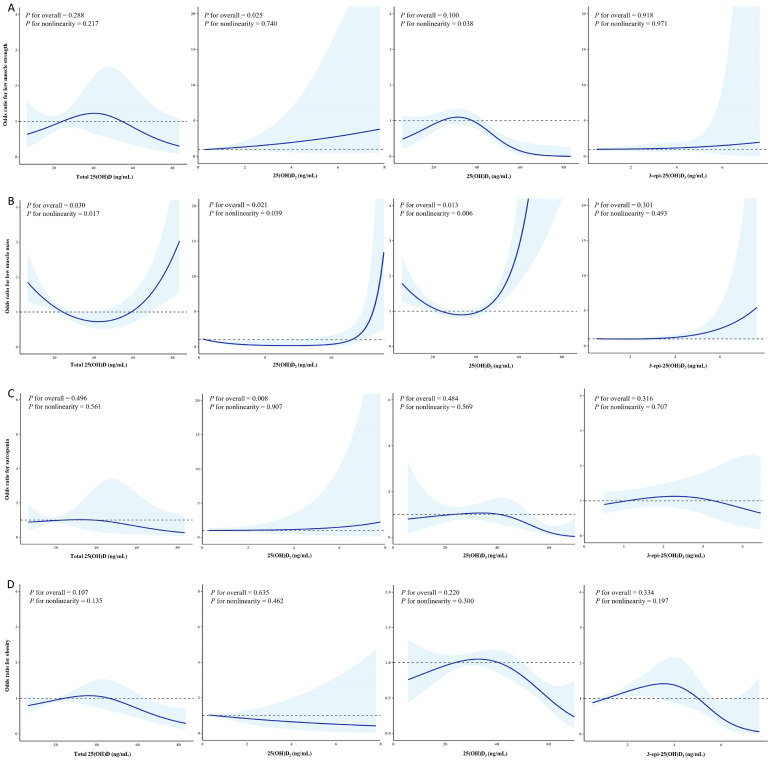
Non-linear relationship between serum 25-hydroxyvitamin D concentrations and body composition phenotypes. Association of total 25(OH)D, 25(OH)D_2_, 25(OH)D_3_, and 3-epi-25(OH)D_3_ levels with low muscle strength (**A**), low muscle mass (**B**), sarcopenia (**C**), and obesity (**D**). Adjusted for age, sex, smoking, alcohol consumption, exercise, household income, energy intake, proportion of macronutrient intake, fiber intake, comorbidities, and seasonal variation in vitamin D status. The shaded areas indicate the 95% confidence intervals. 25(OH)D, 25-hydroxyvitamin D; 25(OH)D_2_, 25-hydroxyvitamin D_2_; 25(OH)D_3_, 25-hydroxyvitamin D_3_; 3-epi-25(OH)D_3_, 3-epi-25-hydroxyvitamin D_3_.

**Table 1 nutrients-17-03013-t001:** Characteristics of the study participants according to vitamin D status.

	Total	Vitamin D Status		*p*
		Deficient	Non-Deficient	
*N*	2612	811	1801	-
Total 25(OH)D (ng/mL)	24.2 ± 0.2	15.8 ± 0.1	28.9 ± 0.2	<0.001
25(OH)D_2_ (ng/mL)	0.5 ± 0.01	0.5 ± 0.01	0.6 ± 0.02	<0.001
25(OH)D_3_ (ng/mL)	23.7 ± 0.2	15.3 ± 0.1	28.3 ± 0.2	<0.001
3-epi-25(OH)D_3_ (ng/mL)	1.1 ± 0.01	0.7 ± 0.01	1.3 ± 0.02	<0.001
Age (years)	52.2 ± 0.7	47.7 ± 1.0	54.6 ± 0.7	<0.001
Sex (females, %)	51.3	47.0	53.2	0.001
Household income (low, %)	22.4	21.2	22.9	0.600
Current smoking (%)	14.4	18.8	12.4	<0.001
Heavy drinking (%)	28.5	32.9	26.3	0.003
Aerobic exercise (yes, %)	44.9	46.6	44.2	0.200
Resistance exercise (yes, %)	29.0	28.4	29.3	0.803
Energy intake (kcal/day)	1761.9 ± 16.0	1824.3 ± 31.5	1728.8 ± 18.6	0.010
Energy from				
Carbohydrate (%)	60.3 ± 0.3	59.8 ± 0.6	60.5 ± 0.4	0.259
Fat (%)	23.7 ± 0.3	24.2 ± 0.5	23.5 ± 0.3	0.172
Protein (%)	16.0 ± 0.1	16.0 ± 0.2	16.0 ± 0.1	0.957
Fiber intake (g/day)	25.7 ± 0.4	25.2 ± 0.6	26.0 ± 0.4	0.259
Medical history (%)				
Diabetes	15.9	14.9	16.3	0.246
Hypertension	33.7	30.7	35.1	0.130
Dyslipidemia	32.0	27.9	33.9	0.003
Waist circumference (cm)	85.0 ± 0.2	85.9 ± 0.4	84.5 ± 0.3	0.006
Body mass index (kg/m^2^)	24.3 ± 0.1	24.7 ± 0.2	24.2 ± 0.1	0.007
Handgrip strength (kg)	33.6 ± 0.3	34.4 ± 0.4	33.2 ± 0.4	0.021
Fat mass (kg)				
Total	19.3 ± 0.2	20.0 ± 0.3	18.9 ± 0.2	0.002
Trunk	9.9 ± 0.1	10.3 ± 0.2	9.7 ± 0.1	0.002
Limbs	8.2 ± 0.1	8.5 ± 0.1	8.2 ± 0.1	0.003
Lean mass (kg)				
Total	47.6 ± 0.3	48.9 ± 0.4	47.0 ± 0.3	0.001
Trunk	27.9 ± 0.1	28.5 ± 0.2	27.5 ± 0.2	0.001
Limbs	19.8 ± 0.1	20.4 ± 0.2	19.4 ± 0.2	0.001
ASM (kg/m^2^)	7.1 ± 0.02	7.2 ± 0.04	7.1 ± 0.02	0.003
Low muscle strength (%)	7.5	6.2	8.2	0.325
Low muscle mass (%)	19.5	16.9	20.6	0.126
Sarcopenia (%)	4.1	3.3	4.5	0.193
Obesity (%)	36.6	38.7	35.6	0.138

Values are expressed as percentages for categorical variables and means ± standard errors for continuous variables. ASM, height-adjusted appendicular skeletal muscle mass; 25(OH)D, 25-hydroxyvitamin D; 25(OH)D_2_, 25-hydroxyvitamin D_2_; 25(OH)D_3_, 25-hydroxyvitamin D_3_; 3-epi-25(OH)D_3,_ 3-epi-25-hydroxyvitamin D_3_.

**Table 2 nutrients-17-03013-t002:** Association between serum 25-hydroxyvitamin D concentrations and body composition factors.

		Total 25(OH)D		25(OH)D_2_		25(OH)D_3_		3-epi-25(OH)D_3_	
		β	*p*	β	*p*	β	*p*	β	*p*
Unadjusted	Waist circumference (cm)	−0.087	<0.001	−0.002	0.154	−0.085	<0.001	0.001	0.575
	Body mass index (kg/m^2^)	−0.313	<0.001	−0.007	0.051	−0.306	<0.001	−0.004	0.210
	Handgrip strength (kg)	−0.120	<0.001	−0.006	<0.001	−0.114	<0.001	−0.003	0.024
	Fat mass (kg)								
	Total	−0.160	<0.001	−0.002	0.201	−0.157	<0.001	−0.003	0.083
	Trunk	−0.298	<0.001	−0.005	0.126	−0.293	<0.001	−0.006	0.127
	Limbs	−0.325	<0.001	−0.003	0.438	−0.322	<0.001	−0.008	0.055
	Lean mass (kg)								
	Total	−0.147	<0.001	−0.006	<0.001	−0.141	<0.001	−0.004	0.001
	Trunk	−0.268	<0.001	−0.010	0.001	−0.258	<0.001	−0.007	0.002
	Limbs	−0.309	<0.001	−0.012	<0.001	−0.297	<0.001	−0.009	<0.001
	ASM (kg/m^2^)	−1.280	<0.001	0.052	0.001	−1.228	<0.001	−0.031	0.005
Adjusted	Waist circumference (cm)	−0.058	0.011	−0.001	0.459	−0.057	0.013	−0.0004	0.784
	Body mass index (kg/m^2^)	−0.154	0.006	−0.002	0.487	−0.151	0.007	−0.004	0.333
	Handgrip strength (kg)	0.118	0.001	−0.005	0.073	0.123	<0.001	0.004	0.121
	Fat mass (kg)								
	Total	−0.115	0.001	−0.002	0.281	−0.113	0.001	−0.003	0.182
	Trunk	−0.195	0.001	−0.004	0.210	−0.191	0.002	−0.005	0.199
	Limbs	−0.284	<0.001	−0.003	0.421	−0.280	<0.001	−0.007	0.153
	Lean mass (kg)								
	Total	0.062	0.040	−0.001	0.747	0.063	0.038	0.003	0.131
	Trunk	0.104	0.051	−0.0002	0.953	0.105	0.051	0.004	0.239
	Limbs	0.135	0.041	−0.003	0.551	0.138	0.037	0.008	0.064
	ASM (kg/m^2^)	0.195	0.461	−0.014	0.497	0.209	0.432	0.017	0.361

Adjusted for age, sex, smoking, alcohol consumption, exercise, household income, energy intake, proportion of macronutrient intake, fiber intake, comorbidities, and seasonal variation in vitamin D status. ASM, height-adjusted appendicular skeletal muscle mass; 25(OH)D, 25-hydroxyvitamin D; 25(OH)D_2_, 25-hydroxyvitamin D_2_; 25(OH)D_3_, 25-hydroxyvitamin D_3_; 3-epi-25(OH)D_3_, 3-epi-25-hydroxyvitamin D_3_.

**Table 3 nutrients-17-03013-t003:** Association between serum 25-hydroxyvitamin D concentrations and body composition phenotypes.

	Unadjusted	*p*	Model 1	*p*	Model 2	*p*	Model 3	*p*
Total 25(OH)D								
Low muscle strength	1.033 (1.015−1.051)	<0.001	1.009 (0.990−1.029)	0.369	0.989 (0.960−1.019)	0.476	0.988 (0.958−1.019)	0.433
Low muscle mass	1.025 (1.013−1.037)	<0.001	1.008 (0.995−1.021)	0.228	1.009 (0.992−1.026)	0.287	1.009 (0.992−1.026)	0.301
Sarcopenia	1.035 (1.017−1.054)	<0.001	1.009 (0.989−1.029)	0.370	0.985 (0.946−1.025)	0.457	0.985 (0.948−1.025)	0.458
Obesity	0.978 (0.967−0.989)	<0.001	0.981 (0.970−0.993)	0.002	0.987 (0.972−1.002)	0.082	0.989 (0.974−1.004)	0.133
25(OH)D_2_								
Low muscle strength	1.269 (1.081−1.490)	0.004	1.164 (1.014−1.337)	0.031	1.266 (1.056−1.518)	0.011	1.279 (1.055−1.552)	0.013
Low muscle mass	1.175 (1.030−1.339)	0.016	1.101 (0.986−1.230)	0.088	1.116 (0.903−1.378)	0.308	1.128 (0.913−1.393)	0.262
Sarcopenia	1.346 (1.107−1.636)	0.003	1.224 (1.037−1.444)	0.017	1.424 (1.166−1.739)	0.001	1.418 (1.162−1.732)	0.001
Obesity	0.948 (0.838−1.073)	0.400	0.975 (0.872−1.090)	0.653	0.922 (0.743−1.143)	0.456	0.942 (0.759−1.169)	0.589
25(OH)D_3_								
Low muscle strength	1.030 (1.011−1.049)	0.002	1.007 (0.987−1.027)	0.516	0.986 (0.957−1.017)	0.369	0.985 (0.955−1.016)	0.333
Low muscle mass	1.024 (1.012−1.036)	<0.001	1.007 (0.994−1.020)	0.294	1.009 (0.991−1.026)	0.329	1.008 (0.991−1.025)	0.346
Sarcopenia	1.030 (1.012−1.049)	0.001	1.004 (0.985−1.024)	0.650	0.978 (0.937−1.020)	0.294	0.978 (0.939−1.019)	0.295
Obesity	0.978 (0.967−0.989)	<0.001	0.981 (0.970−0.993)	0.002	0.987 (0.972−1.002)	0.091	0.989 (0.974−1.004)	0.143
3-epi-25(OH)D_3_								
Low muscle strength	1.490 (1.199−1.850)	<0.001	1.247 (0.949−1.640)	0.113	1.073 (0.696−1.653)	0.749	1.045 (0.681−1.604)	0.839
Low muscle mass	1.159 (1.001−1.343)	0.049	0.999 (0.854−1.170)	0.993	1.121 (0.896−1.404)	0.317	1.166 (0.935−1.454)	0.173
Sarcopenia	1.321 (1.070−1.632)	0.010	1.048 (0.805−1.363)	0.729	0.801 (0.462−1.391)	0.431	0.796 (0.469−1.350)	0.397
Obesity	0.949 (0.824−1.092)	0.464	0.970 (0.839−1.122)	0.683	1.004 (0.838−1.202)	0.967	0.961 (0.798−1.157)	0.675

Values are expressed as odds ratios (95% confidence intervals). Model 1: adjusted for age and sex; Model 2: adjusted for Model 1 + smoking, alcohol consumption, physical activity, household income, energy intake, proportion of macronutrient intake, and fiber intake; Model 3: adjusted for Model 2 + comorbidities and seasonal variation in vitamin D status. 25(OH)D, 25-hydroxyvitamin D; 25(OH)D_2_, 25-hydroxyvitamin D_2_; 25(OH)D_3_, 25-hydroxyvitamin D_3_; 3-epi-25(OH)D_3_, 3-epi-25-hydroxyvitamin D.

## Data Availability

The KNHANES is an open access resource, and the researchers can apply to use the dataset by registering and applying at https://knhanes.kdca.go.kr/knhanes/rawDataDwnld/rawDataDwnld.do (accessed on 19 May 2025). The analytic code described in this manuscript is available upon request.

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
