# Peer review of "Differential Associations of Vitamin D Metabolites with Adiposity and Muscle-Related Phenotypes in Korean Adults: Results from KNHANES 2022–2023"

_nutrients, 2025, doi:10.3390/nu17183013_

Round 1
Reviewer 1 Report
Comments and Suggestions for Authors
The authors have used KNHANES data from 2022 and 2023 to investigate the association between the serum concentration of various types of vitamin D and body composition parameters and related phenotypes in over 2,600 Korean adults aged 20 years or older. The results showed interesting associations, including inverse associations, between, on the one hand, 25(OH)D, 25(OH)D2, 25(OH)D3, and, on the other hand, hand grip strength and lean body/muscle mass. The epimer 3-epi-25(OH)D3 did not show any such significant association.
There are currently three weaknesses in this submission.
First, the intake of vitamin D is a confounder. There is no attempt to estimate this. I note what the authors say in lines 319-21. However, my understanding is that the 2025 KNHANES includes survey data on dietary supplement use; the frequency of eating meat, fish, eggs and dairy products; the names and amounts of food consumed during 24 hours; and the names and amounts of ingredients of home-cooked dishes. Thus, an estimation should be possible.
Second, the authors rightly point out the importance of being able to use LC-MS/MS to differentiate accurately between 25(OH)D3 and its epimers. However, just one sentence (lines 110-3) is devoted to this LC-MS/MS methodology. In order to allow replication of the present findings by other groups, it is important that further pertinent details of the use of this technique be given. I presume that these details are available or can be sought from the Korea Disease Control and Prevention Agency. Ideally, these should include details of the extraction methodology; the type(s) of separation columns used; the ion pair m/z values monitored; the flow rate, solvents, gradient, stop time and column temperature; assay performance parameters; and the validation data (e.g. radioimmunoassay and/or deuterated molecules).
Third, many of the authors' key conclusions are based on the their multiple logistic analyses. However, the authors do not mention sufficient details to allow their analyses to be replicated. At the very least, it would be good if the method of entry of the dependent variables were to be stated please.
Author Response
There are currently three weaknesses in this submission.
First, the intake of vitamin D is a confounder. There is no attempt to estimate this. I note what the authors say in lines 319-21. However, my understanding is that the 2025 KNHANES includes survey data on dietary supplement use; the frequency of eating meat, fish, eggs and dairy products; the names and amounts of food consumed during 24 hours; and the names and amounts of ingredients of home-cooked dishes. Thus, an estimation should be possible.
Answer) Thank you for your valuable comment. As the reviewer noted, vitamin D intake can be estimated in KNHANES using the 24-h dietary recall method and the Korean Nutrition Society database. Indeed, KNHANES 2022 reported an average daily intake of 2.9 μg in Korean adults, which corresponds to approximately a 1 ng/mL contribution to serum vitamin D levels — a minimal amount compared to total serum concentrations.
However, information on vitamin D supplementation, which may substantially affect serum vitamin D levels, is not available in the currently released KNHANES dataset. The evaluation of serum vitamin D in relation to supplement use would be valuable, and further studies are warranted to examine the effects of vitamin D supplementation (such as supplementation of D2 vs D3). Importantly, the primary aim of this study was to explore the relationship between serum vitamin D levels and body composition parameters and related phenotypes, rather than to identify dietary or supplementary sources of vitamin D. We believe our results provide a basis for future research to address the reviewer’s suggestion.
Ref) Ten-year trends in dietary vitamin D intake and its food sources among Koreans: based on the 2013–2022 Korea National Health and Nutrition Examination Survey data. Nutr Res Pract. 2025 Aug;19(4):577-590
Second, the authors rightly point out the importance of being able to use LC-MS/MS to differentiate accurately between 25(OH)D3 and its epimers. However, just one sentence (lines 110-3) is devoted to this LC-MS/MS methodology. In order to allow replication of the present findings by other groups, it is important that further pertinent details of the use of this technique be given. I presume that these details are available or can be sought from the Korea Disease Control and Prevention Agency. Ideally, these should include details of the extraction methodology; the type(s) of separation columns used; the ion pair m/z values monitored; the flow rate, solvents, gradient, stop time and column temperature; assay performance parameters; and the validation data (e.g. radioimmunoassay and/or deuterated molecules).
Answer) As recommended, we have added the content on 2.2. Measurement of serum 25-hydroxyvitamin D concentrations in Materials and Methods section.
Third, many of the authors' key conclusions are based on the their multiple logistic analyses. However, the authors do not mention sufficient details to allow their analyses to be replicated. At the very least, it would be good if the method of entry of the dependent variables were to be stated please.
Answer) As recommended, we have added the content on 2.6. Statistical analysis and 2.4. Definition of body composition phenotypes in Materials and Methods section.

Reviewer 2 Report
Comments and Suggestions for Authors
This very well written manuscript reports a wide range of physiological measurements related to vitamin D status, in a large population of adults throughout South Korea from data obtained in KNHANES survey in 2022 and 2023. The findings are well aligned with other surveys that report that adiposity and muscle strength are negatively associated with the serum concentration of 25(OH)D. An unpredicted and surprising finding from this survey was the positive association of serum 25(OH)D2 concentration with low muscle strength and with sarcopenia.
Overall, this is a very useful contribution to the international range of surveys linking vitamin D status to various physiological characteristics. The finding of associations with the concentration of 25(OH)D2 is entirely new. It would be helpful if the authors, in the Discussion section, could suggest some possibilities to explain this novel finding. The faster rate of decline in the concentration of serum 25(OH)D2 than that of 25(OH)D3 may lead to some hypotheses to be tested for these differences in the associations with these two forms of the 25-hydroxy metabolite of vitamin D. Some suggestions could well be included in the Discussion paragraph from line 280 to 296.
There is one small error that is expressed in many places throughout the text. That is a reference to “serum vitamin D levels”. The circulating blood contains both vitamin D and its metabolite, 25(OH)D. Therefore, it is a mistake to refer to serum vitamin D levels when only the concentration of 25(OH)D was determined. This misleading error is expressed at lines: 200, 205, 215, 225, 229, 232, 245, 249, 258, 312 and 321. This error can readily be corrected.
Author Response
Overall, this is a very useful contribution to the international range of surveys linking vitamin D status to various physiological characteristics. The finding of associations with the concentration of 25(OH)D2 is entirely new. It would be helpful if the authors, in the Discussion section, could suggest some possibilities to explain this novel finding. The faster rate of decline in the concentration of serum 25(OH)D2 than that of 25(OH)D3 may lead to some hypotheses to be tested for these differences in the associations with these two forms of the 25-hydroxy metabolite of vitamin D. Some suggestions could well be included in the Discussion paragraph from line 280 to 296.
Answer) Thank you for your valuable comment. As recommended, we have added the content to the third paragraph in the Discussion section.
There is one small error that is expressed in many places throughout the text. That is a reference to “serum vitamin D levels”. The circulating blood contains both vitamin D and its metabolite, 25(OH)D. Therefore, it is a mistake to refer to serum vitamin D levels when only the concentration of 25(OH)D was determined. This misleading error is expressed at lines: 200, 205, 215, 225, 229, 232, 245, 249, 258, 312 and 321. This error can readily be corrected.
Answer) As recommended, we have changed “serum vitamin D levels” to “serum 25-hydroxyvitamin D concentrations” properly.
